Subject Category:
Biology (whole organism)

Subject Areas:
behaviour/ecology/evolution

Keywords:
aposematism, warning signal, chemical defence, metabolomics

Author for correspondence:
Jingchun Li
e-mail: jingchun.li@colorado.edu

# Brightly coloured tissues in limid bivalves chemically deter predators

Lindsey F. Dougherty[1,2,3], Alexandria K. Niebergall[3], Corey D. Broeckling[4], Kevin L. Schauer[5] and Jingchun Li[1,2]

[1]Department of Ecology and Evolutionary Biology, and [2]Museum of Natural History, University of Colorado Boulder, Boulder, CO, USA
[3]Department of Integrative Biology, University of California Berkeley, Berkeley, CA, USA
[4]Proteomics and Metabolomics Facility, Colorado State University, Fort Collins, CO, USA
[5]Genome Center of Wisconsin, University of Wisconsin Madison, Madison, WI, USA

JL, 0000-0001-7947-0950

Members of the marine bivalve family Limidae are known for their bright appearance. In this study, their colourful tissues were examined as a defence mechanism towards predators. We showed that when attacked by the peacock mantis shrimp (*Odontodactylus scyllarus*), the 'disco' clam, *Ctenoides ales*, opened wide to expose brightly coloured tissues to the predator. The predator also significantly preferred to consume the internal, non-colourful clam tissues than the external, colourful tissues. Mass spectrometry-based metabolomic analysis confirmed that colourful tissues had significantly different chemical compositions than the non-colourful ones. The internal, non-colourful tissues had metabolite profiles more similar to an outgroup bivalve than to the species' own colourful external tissues. A number of the compounds that differentiated the colourful tissues from the non-colourful tissues appeared to be peptide-like, which potentially serve as the underlying defensive compounds. This is the first study demonstrating that colourful bivalve tissues are used for chemical defence.

## 1. Introduction

Chemical defences play important roles in shaping predator–prey interactions [1]. They can influence organisms' behaviours, physiologies and life-history strategies [2,3]. Animals can use diverse chemical compounds for defence—some are taken from the environment (exogenous) and others are synthesized by the organisms (endogenous) [4]. The endogenous defence compounds,

**Figure 1.** Limid bivalves *Ctenoides ales* (*a*) and *Ctenoides scaber* (*b*) exhibit reddish mantle and tentacle tissue derived from carotenoids.

such as peptides, usually fall outside of 'primary' metabolic pathways and are thus termed secondary or specialized metabolites.

Organisms that use noxious chemicals sometimes exhibit bright colours termed aposematism [5]. Aposematic coloration accurately communicates the presence of defensive chemistry and is seen in diverse marine organisms. For example, many unpalatable nudibranchs use bright colours as warning signals [6], and their colourful tissues contain significantly more noxious compounds than the non-colourful, internal tissues [7].

Chemical defences are scarcely reported in marine bivalves. Most species avoid predation by developing thick calcium carbonate shells, occasionally through nest building or soft tissue autotomy [8]. Some species in the families Galeommatidae and Limidae are thought to use endogenous chemicals to deter predators, although their tissue noxiousness has never been confirmed [9,10].

Aposematic displays are also rarely known in bivalves. However, some species in the family Limidae do exhibit coloration derived from carotenoids [11] that appears bright reddish to the human eye (figure 1). In one species, the 'disco clam' *Ctenoides ales* [12], the bright coloration is augmented by a flashing display. Behavioural studies show that the rate of flashing increases significantly in the presence of predators [13]. This suggests that the flashing may augment coloration to aid in conspicuousness as an aposematic signal [14]. To thoroughly investigate this hypothesis, the first step is to test whether the colourful tissues in *C. ales* can actually deter predators and contain noxious chemicals.

In this study, we conducted behavioural trials to determine whether brightly coloured tissues of *C. ales* were distasteful to predators. We also conducted non-targeted metabolomic analysis to compare chemical compositions of its colourful and non-colourful tissues.

## 2. Material and methods

### 2.1. Predator trials

Natural predators of *C. ales* are currently unknown. Therefore, we used a common bivalve predator, the peacock mantis shrimp (*Odontodactylus scyllarus*), which co-occurs with *C. ales* to conduct the experiments. In the first experiment, a live *C. ales* clam (*n* = 7) was presented to a live peacock mantis shrimp predator (*n* = 7), which had been fasted for 2–3 days. The clam (initially closed) was placed less than or equal to 5 cm directly in front of the burrow of *O. scyllarus* during its regular feeding time (12.00–14.00). Interactions were filmed for 5 min to record whether the clam opened to show brightly coloured tissues and the predator's reactions. As a control, the same procedure was applied to the palatable Manila clam *Venerupis philippinarum* (*n* = 7).

In the second experiment, *O. scyllarus* was given a food choice stick with *C. ales* mantle and adductor tissues (less than or equal to 1 cm²) skewered 5 cm apart (*n* = 7). The mantle tissue is colourful and exposed to predators when the clam opens, therefore hypothesized to be noxious. The adductor

**Table 1.** The four bivalve species and tissue types used in metabolomic analysis. Red tissues in *Ctenoides* were assigned as noxious, all others were assumed non-noxious.

| species | mantle tissue | tentacle tissue | gill tissue | adductor muscle |
|---|---|---|---|---|
| *Ctenoides ales* ($n = 5$) | noxious | | | non-noxious |
| *Ctenoides scaber* ($n = 10$) | noxious | | | non-noxious |
| *Argopecten irradians* ($n = 5$) | non-noxious | | | |
| *Spondlyus sp.* ($n = 1$) | non-noxious | | | |

muscle is white and internal—hypothesized to be non-noxious. The choice stick was placed in the water 10 cm directly in front of the burrow of *O. scyllarus*. The left–right presentation of the two tissue types was alternated with every feeding. All predator choices were recorded for 15 min. The predators could consume one type of tissue or both. A Fisher's exact test was used to compare frequencies of consumption of the two tissue types. The same behavioural trials were repeated in the dark ($n = 11$), where the tissue colours were not visible; and then repeated in the light, but with both tissue types dyed using red food colouring ($n = 12$).

In the third experiment, *O. scyllarus* was given a food choice stick with the non-limid bivalve *V. philippinarum* mantle tissue (non-noxious) on one side, and *C. ales* mantle tissue on the other side ($n = 7$). All other conditions were kept the same as the second experiment.

The first, third and the regular choice portion of the second experiment were conducted on seven different mantis shrimps. Each mantis shrimp was only used once in each trial, therefore no learning should occur, as we did not intend to test whether the predators can learn the implications of red colours. The dark treatment and dye-food treatment in the second experiment were conducted on a different set of four mantis shrimps, each used two to three times in the trials. See electronic supplementary material, file S1 for details of specimen supply and voucher information.

## 2.2. Metabolomics

Four bivalve species were used to compare chemical compositions of different tissues through metabolomic analysis. Those included two species from the Limidae family: *C. ales* ($n = 5$) and *Ctenoides scaber* ($n = 10$); one from the Spondylidae family (thorny oysters): *Spondylus* sp. ($n = 1$); and one from the Pectinidae family (scallops): *Argopecten irradians* ($n = 5$). Thorny oyster and scallop species were chosen as control groups because they are closely related to Limidae and are not known to be noxious. Sample sizes were determined by specimen availability.

For each specimen, four types of tissues were extracted: mantle, tentacle, gill and adductor muscle (table 1). Chemical compositions of the samples were analysed at the Proteomics and Metabolomics Facility at Colorado State University using reverse-phase ultra-high-pressure liquid chromatography (UHPLC) coupled to a time of flight (TOF) mass spectrometer (MS) using a stacked injection approach to maximize metabolic coverage (method details in electronic supplementary material, file S1).

For each sample, distinct compounds were identified based on retention time and mass ($m/z$) spectra. Compounds were annotated based on spectral matching to metabolite databases and computational interpretation of spectra. Compounds were categorized as potential peptides if they were observed at a charge state greater than +1. Each compound's signal intensity in every tissue type was calculated and normalized among tissues. *T*-tests were used to determine which compounds were significantly different between *Ctenoides* red tissues and all other non-noxious tissues. A principal component analysis (PCA) and a hierarchical clustering were used to explore overall chemical compositions of different tissue types and bivalve species (method details in electronic supplementary material, file S1).

# 3. Results

## 3.1. Behaviour trials

In the first experiment, *C. ales* opened in 57% of trials after being hit by the predator *O. scyllarus*. The mantis shrimp reacted to physical contact with the bright external tissues with excessive mouthpart cleaning, and

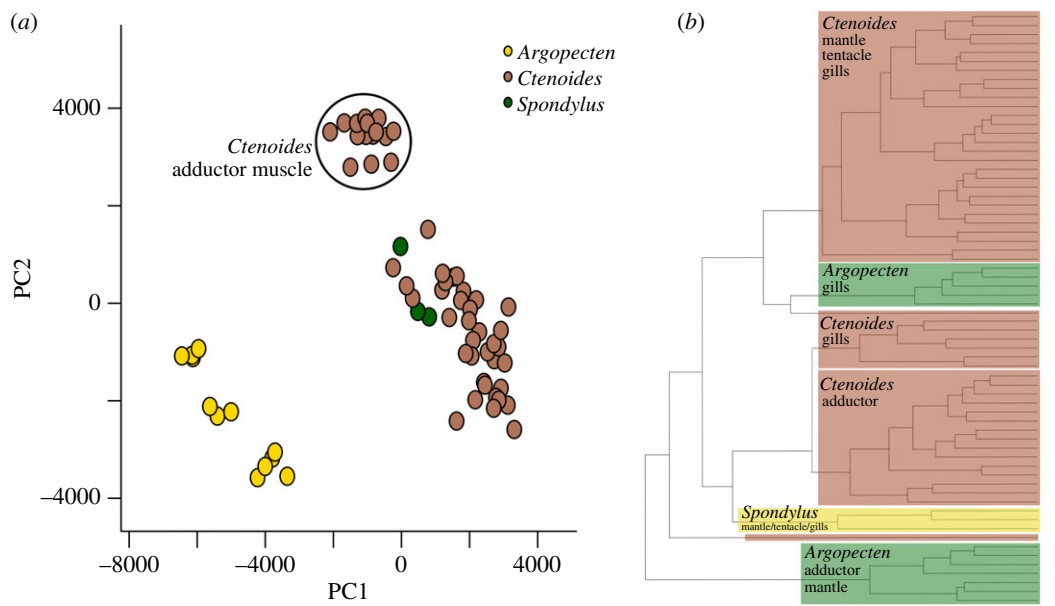

**Figure 2.** (*a*) Scatter plot of the first two principal components of the overall chemical compositions of different bivalve genera. *Argopecten* showed a clear separation from *Ctenoides* and *Spondylus*, and *Ctenoides* adductor muscle showed a clear separation from the remaining *Ctenoides* tissues (mantle, tentacle, gills). (*b*) Hierarchical clustering of the chemical compositions of the three bivalve genera. *Ctenoides* adductor muscle was compositionally closely related to some *Ctenoides* gills and *Spondylus* tissues.

then entered a catatonic state for up to 15 min (electronic supplementary material, file S2); while the palatable bivalve *V. philippinarum* never opened and was cracked and consumed by the mantis shrimp. In the second experiment, *O. scyllarus* initially handled both tissue types, and then consumed one or both types. In total, *O. scyllarus* consumed *C. ales* adductor tissue in 100% of trials, which was significantly higher than the consumption of the mantle tissue in 43% of trials ($n = 7$, $p = 0.03$). Similarly, in the dark trials, *O. scyllarus* consumed the adductor tissue in 100% of trials, and the mantle tissue in 45% of trials ($n = 11$, $p = 0.006$). A similar pattern was found in the trials where the adductor muscles were coloured red; consumption rates were 92% for adductor tissue and 42% for mantle tissue ($n = 12$, $p = 0.01$). In the third experiment, *O. scyllarus* consumed the mantle tissue of the non-limid bivalve *V. philippinarum* in 86% of trials, but only consumed *C. ales* mantle tissue in 29% of trials ($n = 7$, $p = 0.05$).

## 3.2. Metabolomics

Metabolomic analysis detected 1460 compounds, 281 were assigned tentative annotations through structure and spectral database searching. *T*-tests revealed that 809 compounds had significantly higher concentrations in tissues pre-identified as noxious (electronic supplementary material, file S3). Phospholipids, carotenoids and peptides are the most common classes of those annotated compounds showing differential abundance.

In the PCA analysis, *Argopecten* (scallops) showed clear separation from *Ctenoides* and *Spondylus* (thorny oyster) tissues (figure 2*a*). Within *Ctenoides*, the adductor muscle was clearly separated from gill, mantle and tentacle tissues. In the hierarchical clustering analysis, the *Ctenoides* adductor muscle was compositionally more closely related to some *Ctenoides* gills and *Spondylus* than the rest of the *Ctenoides* tissues (figure 2*b*). Given that bivalve gills are used to process food particles, the chemical compositions of gill tissues may not accurately reflect metabolites of the bivalve themselves but instead include contaminants from trapped food particles. If gill tissues were removed from the clustering, it became more obvious that *Ctenoides* adductor muscle is grouped with *Spondylus* tissues.

## 4. Discussion

This is the first study to demonstrate that bivalve tissues can chemically deter predators. Marine predators, such as shrimps and fish, can detect prey metabolites through 'tactile' forms of olfaction [15]. The use of mouthparts as 'aquatic noses' could explain the aversive behaviours of the peacock mantis shrimp after tactile interaction with external colourful disco clam tissues. The food choice trials also demonstrated

that the mantis shrimp significantly prefers to consume the internal, white adductor muscle than the external, colourful mantle. This is not because bivalve mantle tissue, in general, is less preferred, as the mantis shrimp had no trouble consuming mantle tissue of the non-limid Manila clam, supporting the hypothesis that the external tissue of the disco clam is distasteful. Although the tissue is noxious, it does not seem to cause lethal effects on the predators. Live clams cause the predator to enter a catatonic state, but dissected external tissues alone do not. This suggests the noxious chemicals may have short half-lives, can be washed away after detachment or were actively synthetized by the live clams during the attack.

The dark and food-colouring trials showed that the red coloration alone did not deter predators. However, we cannot rule out the colour or disco clam flashing as aposematic signals, as other predators, such as fish, may react differently to the signal. Also, the mantis shrimps used were naive predators that had not interacted with the clams before, therefore may not be familiar with the clams' visual signals. It is possible that the shrimps will learn to avoid the clams based on coloration or flashing over time. In addition, when the initially closed disco clams were attacked, they opened in more than half of the trials to expose the brightly coloured tissue. This is a highly unusual behaviour for bivalves and was not observed in the control group (Manila clams), suggesting that the display may be used as a warning before and during attacks. Overall, we have successfully demonstrated that the red tissues in *Ctenoides* clams are distasteful to predators. But further investigations are needed to test whether they play any aposematic roles.

The metabolomic analysis revealed that the *Ctenoides* colourful tissues were indeed chemically very different from the internal adductor muscles. The adductor in fact shares more metabolomic similarities with the edible thorny oyster outgroup. More than half of the detected compounds have significantly higher concentrations in the noxious tissues and many could not be annotated, making it difficult to pinpoint specific chemicals that are responsible for the distastefulness. But, some compounds can probably be excluded. Certain phospholipids and carotenoids are found in higher concentrations in the noxious tissues, but both are generally non-noxious. The high levels of carotenoids are not surprising, as they are responsible for the tissues' bright coloration [13]. Other promising candidates for the noxious chemicals are peptides, as they were some of the most significant compounds differentiating noxious from non-noxious tissues. Many marine peptides are bioactive compounds and have cultivated a great deal of interest due to their antihypertensive, antioxidative, anticoagulant and antimicrobial components [16]. Further analyses to characterize these peptides are being conducted.

This study is a pioneer analysis on bivalve chemical defence. It demonstrated the complexity of chemical defence in marine invertebrates and the use of metabolomics as a tool to explore potential bioactive compounds. Even the most comprehensive metabolomic structure databases offer poor coverage of marine natural products. The wealth of data generated in this work provides ample opportunities for in-depth analyses on these compounds and paves the way for establishing marine metabolomic databases.

Ethics. Animals used in the study were purchased from aquarium stores. The use of invertebrate animals does not require protocol approval by the Institutional Animal Care and Use Committee (IACUC), University of Colorado Boulder.

Data accessibility. Data are provided in the electronic supplementary material, table.

Authors' contributions. L.F.D. designed and performed the behavioural study, prepared specimens for metabolomics, participated in data analysis and drafted the manuscript; A.K.N. performed the behavioural study; C.D.B. carried out the metabolomic and statistical analysis; K.L.S. designed the metabolomic analysis and helped with the statistical analysis; J.L. designed the behavioural study, helped with statistical analysis, wrote the manuscript and coordinated the study. All authors gave approval for publication.

Competing interests. The authors have no competing interests.

Funding. This study is supported by internal funds to J.L. from the Museum of Natural History, University of Colorado Boulder.

Acknowledgements. We thank Dr Roy Caldwell, Jade Sigler, Stine Skalmerud and Brenna Swafford for aid with behavioural trials.

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
