## [Reviewer comments · Royal Society Open Science]

Review History

RSOS-191298.R0 (Original submission)

Review form: Reviewer 1

Is the manuscript scientifically sound in its present form?

Yes

Are the interpretations and conclusions justified by the results?

No

Is the language acceptable?

Yes

Do you have any ethical concerns with this paper?

No

Have you any concerns about statistical analyses in this paper?

No

Recommendation?

Accept with minor revision (please list in comments)

Comments to the Author(s)

It seems the authors have not addressed any of mine or reviewer 3's comments from the last review, not even the correction of typos that were pointed out. So, my previous comments remain. I would also like for the authors respond to reviewer 3's specific comments as they point out several things that need clarification that I missed.

In addition to my previous comments:

Reviewer 3 asks whether the same predators were used across trials. This information is of particular interest for the second experiment where mantis shrimp are first offered a choice of the two tissue types unmanipulated, then offered the same tissues but in the dark (without visual cues), and finally offered the two tissues dyed the same red colour. If the same mantis shrimp are used across each of these trials, this would show that they did not learn to avoid the red colour (lending no support for aposematism). But I am thinking that the first part of this experiment (n=7) was run at the same time as the previous experiment, whereas parts 2 and 3 (n=11, n=12) were run later and possibly with different mantis shrimp individuals? If so, the authors did not test whether the mantis shrimp learned to avoid the red colour and the role of the red colour is still potentially aposematic for the mantis shrimp (and other predators) but simply not tested yet. You would not expect the red colouration to deter naïve predators on its own (unless it is an innate aversion) until they had learned that it is associated with unpalatable tissues. If the experiments were run at different times of year and with different individuals this should be clearly stated. And if parts 2 & 3 of the experiment were run with different individuals, it would also be helpful to be very clear for the reader that you are testing whether the red tissues are unpalatable compared to other tissues, but that your experiments are not designed to test whether the red colouration is aposematic.

Previous comments from myself and reviewer 3:

Reviewer(s)' Comments to Author:

Referee: 1

Comments to the Author(s)

This paper is greatly strengthened with the addition of the two new experiments, and I am now satisfied with the author's conclusion. I believe they have sufficiently disentangled colouration and chemical defense. I would still like some clarification regarding the first experiment with the live disco clam and the new palatable bivalve control. Otherwise, I have only a few suggestions for minor changes below.

Lines 32 -34: I think the abstract would read better with consistent terminology for the tissues. For instance saying "colourful tissues" and "non-colourful tissues" as in the previous sentence rather than "tissues pre-identified as noxious" and "tissues pre-identified as non-noxious." It seems odd to use these terms in the abstract since they are not used elsewhere in the text (except line 125 referring to the ESM file).

Methods: Please mention here that there is an ESM file with the collection information for the mantis shrimp and clams used in the experiments.

Line 113: If *V. philippinarum* was attacked and eaten by the mantis shrimp please say so in the text. The way it is presented it seems *V. philippinarum* was ignored by the mantis shrimp, that

the internal tissues were never touched, and that (like the disco clam) it was also not consumed, which does not really provide an adequate control. If the clams were attacked, this control is suitable for demonstrating that the disco clam opened more frequently than another bivalve species. However, for comparing predator behavior, I would have liked to see a comparison between successful predation attempts and predation attempts that were unsuccessful due to chemical defense. If *V. philippinarum* was not eaten, I would suggest clarifying the purpose of this control in the text. If you have videos of the control experiment with the mantis shrimp attacking and eating *V. philippinarum* it might be useful to add this to the ESM along with the predation attempt with the disco clam for the purpose of comparison. This way the reader can better understand what is meant when you say that the mantis shrimp did not react abnormally.

Line 118: Typo. I suggest replacing “Same patterns were” with “A similar pattern was”

Referee: 3

Comments to the Author(s)

This study investigates the defensive function of the red mantle tissue of the mussel *Ctenoides ales*. The authors conducted a set of behavioural experiments to investigate predators', peacock mantis shrimps, response to the tissue. First, the predators were given a choice between the mantle and adductor tissues of *Ctenoides ales*, and between the mantle and adductor tissues of another, palatable mussel species. This was repeated in the dark to conceal visual differences between the tissues. Finally, the predators were given a choice between the mantle tissues of *Ctenoides ales* and another, non-noxious mussel species, after the tissues had been dyed red to mask any differences in tissue conspicuousness. These choice experiments suggest that the mantle tissue of *C. ales* was distasteful to the predators. The presence or absence of the visual cues did not influence this outcome, suggesting that the avoidance response was induced by the chemical properties and not by the visual properties of the tissue. Hence, the results do not lend support to the hypothesis that the red mantle tissue would have a protective function based on aposematism (but it is important to bear in mind that the tissue may provide aposematic protection against other predators with different vision, such as fishes). A PCA based on metabolomics analysis showed differences in the chemical composition of different parts of *C. ales* as well as other mussel species. However, to me it seems to emphasise the difference of the abductor muscle from other tissue types and tells little about which chemical properties are distinct to the mantle tissue of *C. ales* and contribute to its distastefulness.

I appreciate the use of two different ways to test the signalling function of the red mantle tissue in the experimental part of the study, but I also have some concerns regarding the manuscript. My main concern is that I think that the findings are interesting mainly from a taxon-specific point of view, but they contribute little to the broader theoretical framework of prey defences.

Detailed comments

The title: the emphasis on bright colour is not in agreement with your results showing no effect of colour

L. 46 and elsewhere: Bright coloration/appearance and colourfulness are words that have been used a lot in the text. Yet, the concept is very subjective and fuzzy. What is experienced as bright or colourful depends on the visual system of the viewer and on the visual environment.

Methods: I would like to see some sentences about the biology of the study species, such as its distribution area and its natural predators. Also, please explain why the peacock mantis shrimp was chosen as the predator for the experiments.

L. 70: Explain “regular feeding time”. Maybe the supplementary method section could present more details also about the behavioural experiments.

L. 68-73: Were same predators used in several experiments or were all predators used only once?

L. 80: Change “trails” to “trials”.

L. 116-120: I suggest that you include the sample when you report the results of the Fisher’s exact tests. (It was confusing to see that 43% yielded $P=0.03$, but 45% yielded $P=0.006$).

L. 138: Again, this appears misleading. It was not the colour that deterred the predators.

L. 151: Remember also that other potential predators of the mussel may have different visual system or they may show different innate avoidance response. In other words, the mantle tissue may have an aposematic function towards some other predator.

Review form: Reviewer 2

Is the manuscript scientifically sound in its present form?

Yes

Are the interpretations and conclusions justified by the results?

Yes

Is the language acceptable?

Yes

Do you have any ethical concerns with this paper?

No

Have you any concerns about statistical analyses in this paper?

No

Recommendation?

Accept with minor revision (please list in comments)

Comments to the Author(s)

To the Authors,

I appreciate the changes that the authors have made to the manuscript since it was last in review. In particular I believe the new behavioural trials have filled some important gaps in the experimental framework.

I have a few comments outlined line by line below:

Line 46: Replace ‘colorations’ with ‘colors’

Line 49: Again, replace ‘colorations’ with ‘colors’

Lines 74-85: Were these three experiments conducted on the same seven mantis shrimp?

Either way I think this should be stated here

Lines 76- 77: Replace ‘,’ with ‘.’

Lines 77-78: Was the left-right presentation order randomised or controlled on the choice stick?

Lines 80-82: I think these two control experiments are a very nice touch

Lines 115-121: When reporting percentages, perhaps it's worth reminding the reader of the sample size

Lines 151: You could also mention that the red may act as an aposematic signal to other species of predator, some fish have been shown to be able to learn aposematic signals.

Lines 151-154: Were all mantis shrimp naïve throughout all the experiments, or were the same individuals used for multiple experiments – see my comment for lines 74-85.

Supplementary Methods: The title needs to be updated

Decision letter (RSOS-191298.R0)

22-Aug-2019

Dear Dr Li

On behalf of the Editors, I am pleased to inform you that your Manuscript RSOS-191298 entitled "Brightly-Colored Tissues in Limid Bivalves Deter Predators" has been accepted for publication in Royal Society Open Science subject to minor revision in accordance with the referee suggestions. Please find the referees' comments at the end of this email.

The reviewers and handling editors have recommended publication, but also suggest some minor revisions to your manuscript. Therefore, I invite you to respond to the comments and revise your manuscript.

- Ethics statement

- Data accessibility

If you wish to submit your supporting data or code to Dryad (<http://datadryad.org/>), or modify your current submission to dryad, please use the following link:
<http://datadryad.org/submit?journalID=RSOS&manu=RSOS-191298>

- Competing interests

- Authors' contributions

- Acknowledgements

- Funding statement

Because the schedule for publication is very tight, it is a condition of publication that you submit the revised version of your manuscript before 31-Aug-2019. Please note that the revision deadline will expire at 00.00am on this date. If you do not think you will be able to meet this date please let me know immediately.

on behalf of Kevin Padian (Subject Editor)
openscience@royalsociety.org

Reviewer comments to Author:

Reviewer: 1

It seems the authors have not addressed any of mine or reviewer 3's comments from the last review, not even the correction of typos that were pointed out. So, my previous comments remain. I would also like for the authors respond to reviewer 3's specific comments as they point out several things that need clarification that I missed.

In addition to my previous comments:

Reviewer 3 asks whether the same predators were used across trials. This information is of particular interest for the second experiment where mantis shrimp are first offered a choice of the two tissue types unmanipulated, then offered the same tissues but in the dark (without visual cues), and finally offered the two tissues dyed the same red colour. If the same mantis shrimp are used across each of these trials, this would show that they did not learn to avoid the red colour (lending no support for aposematism). But I am thinking that the first part of this experiment (n=7) was run at the same time as the previous experiment, whereas parts 2 and 3 (n=11, n=12) were run later and possibly with different mantis shrimp individuals? If so, the authors did not test whether the mantis shrimp learned to avoid the red colour and the role of the red colour is still potentially aposematic for the mantis shrimp (and other predators) but simply not tested yet. You would not expect the red colouration to deter naïve predators on its own (unless it is an innate aversion) until they had learned that it is associated with unpalatable tissues. If the experiments were run at different times of year and with different individuals this should be clearly stated. And if parts 2 & 3 of the experiment were run with different individuals, it would also be helpful to be very clear for the reader that you are testing whether the red tissues are unpalatable compared to other tissues, but that your experiments are not designed to test whether the red colouration is aposematic.

Previous comments from myself and reviewer 3:

Reviewer(s)' Comments to Author:

Referee: 1

Comments to the Author(s)

This paper is greatly strengthened with the addition of the two new experiments, and I am now satisfied with the author's conclusion. I believe they have sufficiently disentangled colouration and chemical defense. I would still like some clarification regarding the first experiment with the live disco clam and the new palatable bivalve control. Otherwise, I have only a few suggestions for minor changes below.

Lines 32 -34: I think the abstract would read better with consistent terminology for the tissues. For instance saying "colourful tissues" and "non-colourful tissues" as in the previous sentence rather than "tissues pre-identified as noxious" and "tissues pre-identified as non-noxious." It seems odd to use these terms in the abstract since they are not used elsewhere in the text (except line 125 referring to the ESM file).

Methods: Please mention here that there is an ESM file with the collection information for the mantis shrimp and clams used in the experiments.

Line 113: If *V. philippinarum* was attacked and eaten by the mantis shrimp please say so in the text. The way it is presented it seems *V. philippinarum* was ignored by the mantis shrimp, that the internal tissues were never touched, and that (like the disco clam) it was also not consumed,

which does not really provide an adequate control. If the clams were attacked, this control is suitable for demonstrating that the disco clam opened more frequently than another bivalve species. However, for comparing predator behavior, I would have liked to see a comparison between successful predation attempts and predation attempts that were unsuccessful due to chemical defense. If *V. philippinarum* was not eaten, I would suggest clarifying the purpose of this control in the text. If you have videos of the control experiment with the mantis shrimp attacking and eating *V. philippinarum* it might be useful to add this to the ESM along with the predation attempt with the disco clam for the purpose of comparison. This way the reader can better understand what is meant when you say that the mantis shrimp did not react abnormally.

Line 118: Typo. I suggest replacing "Same patterns were" with "A similar pattern was"

Referee: 3

Comments to the Author(s)

This study investigates the defensive function of the red mantle tissue of the mussel *Ctenoides ales*. The authors conducted a set of behavioural experiments to investigate predators', peacock mantis shrimps, response to the tissue. First, the predators were given a choice between the mantle and adductor tissues of *Ctenoides ales*, and between the mantle and adductor tissues of another, palatable mussel species. This was repeated in the dark to conceal visual differences between the tissues. Finally, the predators were given a choice between the mantle tissues of *Ctenoides ales* and another, non-noxious mussel species, after the tissues had been dyed red to mask any differences in tissue conspicuousness. These choice experiments suggest that the mantle tissue of *C. ales* was distasteful to the predators. The presence or absence of the visual cues did not influence this outcome, suggesting that the avoidance response was induced by the chemical properties and not by the visual properties of the tissue. Hence, the results do not lend support to the hypothesis that the red mantle tissue would have a protective function based on aposematism (but it is important to bear in mind that the tissue may provide aposematic protection against other predators with different vision, such as fishes). A PCA based on metabolomics analysis showed differences in the chemical composition of different parts of *C. ales* as well as other mussel species. However, to me it seems to emphasise the difference of the abductor muscle from other tissue types and tells little about which chemical properties are distinct to the mantle tissue of *C. ales* and contribute to its distastefulness.

I appreciate the use of two different ways to test the signalling function of the red mantle tissue in the experimental part of the study, but I also have some concerns regarding the manuscript. My main concern is that I think that the findings are interesting mainly from a taxon-specific point of view, but they contribute little to the broader theoretical framework of prey defences.

Detailed comments

The title: the emphasis on bright colour is not in agreement with your results showing no effect of colour

L. 46 and elsewhere: Bright coloration/appearance and colourfulness are words that have been used a lot in the text. Yet, the concept is very subjective and fuzzy. What is experienced as bright or colourful depends on the visual system of the viewer and on the visual environment.

Methods: I would like to see some sentences about the biology of the study species, such as its distribution area and its natural predators. Also, please explain why the peacock mantis shrimp was chosen as the predator for the experiments.

L. 70: Explain “regular feeding time”. Maybe the supplementary method section could present more details also about the behavioural experiments.

L. 68-73: Were same predators used in several experiments or were all predators used only once?

L. 80: Change “trails” to “trials”.

L. 116-120: I suggest that you include the sample when you report the results of the Fisher’s exact tests. (It was confusing to see that 43% yielded $P=0.03$, but 45% yielded $P=0.006$).

L. 138: Again, this appears misleading. It was not the colour that deterred the predators.

L. 151: Remember also that other potential predators of the mussel may have different visual system or they may show different innate avoidance response. In other words, the mantle tissue may have an aposematic function towards some other predator.

Reviewer: 2

Comments to the Author(s)

To the Authors,

I appreciate the changes that the authors have made to the manuscript since it was last in review. In particular I believe the new behavioural trials have filled some important gaps in the experimental framework.

I have a few comments outlined line by line below:

Line 46: Replace ‘colorations’ with ‘colors’

Line 49: Again, replace ‘colorations’ with ‘colors’

Lines 74-85: Were these three experiments conducted on the same seven mantis shrimp?

Either way I think this should be stated here

Lines 76- 77: Replace ‘,’ with ‘:’

Lines 77-78: Was the left-right presentation order randomised or controlled on the choice stick?

Lines 80-82: I think these two control experiments are a very nice touch

Lines 115-121: When reporting percentages, perhaps it’s worth reminding the reader of the sample size

Lines 151: You could also mention that the red may act as an aposematic signal to other species of predator, some fish have been shown to be able to learn aposematic signals.

Lines 151-154: Were all mantis shrimp naïve throughout all the experiments, or were the same individuals used for multiple experiments – see my comment for lines 74-85.

Supplementary Methods: The title needs to be updated

Author's Response to Decision Letter for (RSOS-191298.R0)

See Appendix A.

Decision letter (RSOS-191298.R1)

05-Sep-2019

Dear Dr Li,

I am pleased to inform you that your manuscript entitled "Brightly-Colored Tissues in Limid Bivalves Chemically Deter Predators" is now accepted for publication in Royal Society Open Science.

on behalf of Mr Andrew Dunn (Associate Editor) and Kevin Padian (Subject Editor)
openscience@royalsociety.org

Associate Editor Comments to Author (Mr Andrew Dunn):

Associate Editor: 1

Comments to the Author:

(There are no comments.)

Reviewer comments to Author:

Follow Royal Society Publishing on Twitter: [@RSocPublishing](https://twitter.com/RSocPublishing)

Appendix A

Dear Royal Society Open Science editor and reviewers,

My co-authors and I greatly appreciate the constructive comments. We have addressed all of them. Please see below our detailed responses.

Editor

The reviewers and handling editors have recommended publication, but also suggest some minor revisions to your manuscript. Therefore, I invite you to respond to the comments and revise your manuscript.

- Ethics statement

- Funding statement

Response: We have updated the Ethic statement and funding statement:

Ethics

Animals used in the study were purchased from aquarium stores. Use of invertebrate animals does not require protocol approval by the Institutional Animal Care and Use Committee (IACUC), University of Colorado Boulder.

Funding Statement

This study is supported by internal funds to JL from the Museum of Natural History, University of Colorado Boulder

Reviewer: 1

Reviewer 3 asks whether the same predators were used across trials. This information is of particular interest for the second experiment where mantis shrimp are first offered a choice of the two tissue types unmanipulated, then offered the same tissues but in the dark (without visual cues), and finally offered the two tissues dyed the same red colour. If the same mantis shrimp are used across each of these trials, this would show that they did not learn to avoid the red colour (lending no support for aposematism). But I am thinking that the first part of this experiment (n=7) was run at the same time as the previous experiment, whereas parts 2 and 3 (n=11, n=12) were run later and possibly with different mantis shrimp individuals? If so, the authors did not test whether the mantis shrimp learned to avoid the red colour and the role of the red colour is still potentially aposematic for the mantis

shrimp (and other predators) but simply not tested yet. You would not expect the red colouration to deter naïve predators on its own (unless it is an innate aversion) until they had learned that it is associated with unpalatable tissues. If the experiments were run at different times of year and with different individuals this should be clearly stated. And if parts 2 & 3 of the experiment were run with different individuals, it would also be helpful to be very clear for the reader that you are testing whether the red tissues are unpalatable compared to other tissues, but that your experiments are not designed to test whether the red colouration is aposematic.

Response: Thank you for pointing out this topic. Reviewer 1 was right. The first part and the later added experiments were run on different set of individuals. Each different mantis shrimp was exposed to the colorful tissues 1 to 3 times, not enough time for them to learn the colorations. In particular, the dark treatment and dye-food treatment shouldn't facilitate predator learning. Because the mantis shrimps were either kept in dark or the two choices were both red. We deliberately set it up this way, so we are only testing if the tissues are unpalatable, not whether the predators can learn. This goal is stated clearly in the introduction: "In this study, we conducted behavioral trials to determine whether brightly-colored tissues of *Ctenoides ales* were distasteful to predators."

We added the text to make this point more clear:

New in methods: The first, third, and the regular choice portion of the second experiment were conducted on seven different mantis shrimps. Each mantis shrimp was only used once in each trial, therefore no learning should occur, as we did not intent to test whether the predators can learn the implications of red colors. The dark treatment and dye-food treatment in the second experiment were conducted on a different set of four mantis shrimps, each used 2-3 times in the trials. Please see ESM file 1 for details of specimen supply and voucher information.

Lines 32 -34: I think the abstract would read better with consistent terminology for the tissues. For instance saying "colourful tissues" and "non-colourful tissues" as in the previous sentence rather than "tissues pre-identified as noxious" and "tissues pre-identified as non-noxious." It seems odd to use these terms in the abstract since they are not used elsewhere in the text (except line 125 referring to the ESM file).

Response: We have changed the text:

New: Mass spectrometry-based metabolomic analysis confirmed that colorful tissues had significantly different chemical compositions than the non-colorful ones.

Methods: Please mention here that there is an ESM file with the collection information for the mantis shrimp and clams used in the experiments.

Response: This information is added to the text.

New: Please see ESM file 1 for details of specimen supply and voucher information.

Line 113: If *V. philippinarum* was attacked and eaten by the mantis shrimp please say so in the text. The way it is presented it seems *V. philippinarum* was ignored by the mantis shrimp, that the internal tissues were never touched, and that (like the disco clam) it was also not consumed, which does not really provide an adequate control. If the clams were attacked, this control is suitable for demonstrating that the disco clam opened more frequently than another bivalve species. However, for comparing predator behavior, I would have liked to see a comparison between successful predation attempts and predation attempts that were unsuccessful due to chemical defense. If *V. philippinarum* was not eaten, I would suggest clarifying the purpose of this control in the text. If you have videos of the control experiment with the mantis shrimp attacking and eating *V. philippinarum* it might be useful to add this to the ESM along with the predation attempt with the disco clam for the purpose of comparison. This way the reader can better understand what is meant when you say that the mantis shrimp did not react abnormally.

Response: The manila clams were consumed by the mantis shrimps. Unfortunately we didn't take videos of it. But there are plenty of online videos already demonstrating it, such as this one: <https://www.youtube.com/watch?v=i-ahuZEvWH8>. We modified the text to clarify this point:

New: while the palatable bivalve *Venerupis philippinarum* never opened, and was cracked and consumed by the mantis shrimp.

Line 118: Typo. I suggest replacing "Same patterns were" with "A similar pattern was"

Response: We have changed the text to "A similar pattern was".

Referee: 3

Detailed comments

The title: the emphasis on bright colour is not in agreement with your results showing no effect of colour

Response: This particular study is testing whether the colorful tissues are distasteful, not the effect of the color, as emphasized in the introduction: "To thoroughly investigate this hypothesis, the **first step** is to test whether the colorful tissues in *C. ales* **can actually deter predators** and contain noxious chemicals...we conducted behavioral trials to determine whether brightly-colored tissues of *Ctenoides ales* **were distasteful to predators.**". To make this point clear, we

changed the title to “Brightly-Colored Tissues in Limid Bivalves Chemically Deter Predators”

L. 46 and elsewhere: Bright coloration/appearance and colourfulness are words that have been used a lot in the text. Yet, the concept is very subjective and fuzzy. What is experienced as bright or colourful depends on the visual system of the viewer and on the visual environment.

Response: The bright coloration is perceived by the human eye. Due to the scope of this study, we did not model the optic properties of the colored tissue and how it can be interpreted by the predator, because we are merely testing its chemical property. We modified the introduction to make it clear:

Original: However, some species in the family Limidae do exhibit bright reddish coloration (Fig. 1) derived from carotenoids [11].

New: However, some species in the family Limidae do exhibit coloration derived from carotenoids [11] that appears bright reddish to the human eye (Fig. 1).

Methods: I would like to see some sentences about the biology of the study species, such as its distribution area and its natural predators. Also, please explain why the peacock mantis shrimp was chosen as the predator for the experiments.

Response: The biology of limid bivalves is not well studied. It is hard to observe the animals in their natural habitat (reefs) long-term to comprehensively document their predators. In fact, we’ve taken underwater videos of some of the species for days, but no predation events were documented. Therefore, we can only infer that their predators are usual bivalve predators, such as crustaceans, gastropods, and fish. The peacock mantis shrimp was chosen because it co-occurs with *Ctenoides ales* and is known to be a very effective predator. Detailed information on the localities of the bivalves is in ESM file 1. We’ve modified the text to explain why mantis shrimps are used.

New: Natural predators of *Ctenoides ales* are currently unknown. Therefore, we used a common bivalve predator, the peacock mantis shrimp (*Odontodactylus scyllarus*), which co-occurs with *C. ales* to conduct the experiment.

L. 70: Explain “regular feeding time”. Maybe the supplementary method section could present more details also about the behavioural experiments.

Response: Regular feeding time is between 12-2pm. It is added to the text:

New: The clam (initially closed) was placed ≤ 5 cm directly in front of the burrow of *O. scyllarus* during its regular feeding time (12-2 PM).

L. 68-73: Were same predators used in several experiments or were all predators used only once?

Response: Please see our response to reviewer 1's first question.

L. 80: Change "trails" to "trials".

Response: It is changed.

L. 116-120: I suggest that you include the sample when you report the results of the Fisher's exact tests. (It was confusing to see that 43% yielded $P=0.03$, but 45% yielded $P=0.006$).

Response: All sample sizes are now included in the text.

L. 138: Again, this appears misleading. It was not the colour that deterred the predators.

Response: We deleted the word "color".

L. 151: Remember also that other potential predators of the mussel may have different visual system or they may show different innate avoidance response. In other words, the mantle tissue may have an aposematic function towards some other predator.

Response: Thank you for bringing this up. We modified the text to reflect this point.

New: However, we cannot rule out color or disco clam flashing as aposematic signals, as other predators, such as fish, may react differently to the signal.

Reviewer: 2

Line 46: Replace 'colorations' with 'colors'
Line 49: Again, replace 'colorations' with 'colors'

Response: This is fixed.

Lines 74-85: Were these three experiments conducted on the same seven mantis shrimp? Either way I think this should be stated here

Response: Please see our response to reviewer 1's first questions.

Lines 76- 77: Replace ',' with ':'

Response: This is fixed.

Lines 77-78: Was the left-right presentation order randomised or controlled on the choice stick?

Response: The left-right presentation was alternated with every feeding. We added it to the text.

Lines 115-121: When reporting percentages, perhaps it's worth reminding the reader of the sample size

Response: All sample sizes are now included in the text.

Lines 151: You could also mention that the red may act as an aposematic signal to other species of predator, some fish have been shown to be able to learn aposematic signals.

Response: We modified the text to reflect this point.

New: However, we cannot rule out color or disco clam flashing as aposematic signals, as other predators, such as fish, may react differently to the signal.

Lines 151-154: Were all mantis shrimp naïve throughout all the experiments, or were the same individuals used for multiple experiments

Response: Please see our response to reviewer 1's first question.

lines 74-85.
Supplementary Methods: The title needs to be updated

Response: The title is updated.